# Cerebral Vasospasm as a Critical Yet Overlooked Complication Following Tumor Craniotomy: A Systematic Review of Case Reports and Case Series

**DOI:** 10.3390/jcm14072415

**Published:** 2025-04-01

**Authors:** Khairunnisai Tarimah, Dewi Yulianti Bisri, Radian Ahmad Halimi, Elvan Wiyarta

**Affiliations:** 1Department of Anesthesiology and Intensive Therapy Subdivision Neuroanesthesia and Critical Care, Dr Hasan Sadikin Hospital, Padjadjaran University, Bandung 40161, West Java, Indonesia; yuliantibisri@yahoo.com (D.Y.B.); radianhalimi@gmail.com (R.A.H.); 2Department of Anesthesiology and Intensive Therapy, RSUD Kota Mataram, Al-Azhar Islamic University Mataram, Mataram 83127, West Nusa Tenggara, Indonesia; 3Intensive Care Department, University of Indonesia Hospital, Depok 16424, West Jawa, Indonesia; elvan.wiyarta@ui.ac.id; 4Service Department, Risetku, South Jakarta 12820, Jakarta, Indonesia

**Keywords:** brain tumor, craniotomy, resection, tumor, vasospasm, cerebral

## Abstract

**Background:** Cerebral vasospasm after craniotomy tumor (CVACT) is a rare complication that can occur following tumor craniotomy and significantly affects the outcome of patients. Unfortunately, it is not well understood, leading to delayed and ineffective management. This study aims to investigate CVACT by examining the factors contributing to its occurrence, its underlying mechanisms, diagnostic approaches, management strategies, and outcomes. The goal is to identify the characteristics and risk factors associated with CVACT, its clinical symptoms, diagnostic methods, management options, and potential outcomes. **Methods:** A systematic search used relevant keywords to identify cases of “cerebral vasospasm” after tumor resection in PubMed and Science Direct databases. Relevant cross-references were added by manually searching the references of all retrieved articles. **Result:** We included 60 inclusion patients from 14 case reports and 13 case series with 33 (55%) females and 27 (45%) males with a mean age of 44.05 ± 16.8 years. The most common tumors were pituitary adenomas, which were found in 22 (36.66%), the most common tumor location was the middle cranial fossa (75%), and the most common surgery technique used was transsphenoidal surgery (50%). Most of those who experience vasospasm have a craniotomy with the TSS technique (50%) with complications of intraoperative bleeding. The range of onset of VS symptoms postoperatively was 0–30 days (mean 6.59 d). The symptoms included asymptomatic, headache, loss of vision, hemiparesis, diplopia, etc. The vascular involvement was mainly anterior circulation (78.33%). The diagnostic tools most commonly used were angiography and transcranial doppler (TCD). The most common management of VS from the included studies was pharmacology. The survival rate was 61.66%. We found the tumor location and vascular-affected vasospasm were significantly correlated with mortality rates: *p* = 0.015 and *p* = 0.02. **Conclusions:** Cerebral vasospasm after craniotomy tumor removal (CVACT) frequently arises in tumors situated in the medial cranial fossa, predominantly pituitary adenomas and meningiomas. The minimally invasive surgical approach of TSS may contribute to the mechanism of CVACT incidence. The existence of preoperative vascular pathology, as encasement or narrowing, appears to be a predictor alongside the incidence of intra- or postoperative hemorrhage. The vascular structures most susceptible to vasospasm are located in the anterior circulation of the Willis circle, which appears to correlate with the vascular problems that typically undergo preoperative encasement of the internal carotid artery (ICA). The most reliable and real time diagnostic instrument employed is TCD, while imaging continues to be the gold standard. Nimodipine treatment continues to be a viable therapeutic option that can enhance patient outcomes.

## 1. Introduction

Craniotomy tumor removal is a management strategy aimed at reducing intracranial pressure (ICP) resulting from increased intracranial mass and volume in brain tumor patients [1]. Following a tumor craniotomy, patients often require intensive care unit (ICU) treatment due to complications such as intraoperative bleeding, prolonged surgery, hemodynamic instability, or cranial nerve injury [2]. Patients in the ICU may experience neurological deterioration, and obtaining diagnostic imaging can be challenging due to limited facilities or non-transportable patients.

Cerebral vasospasm (CV) is a condition characterized by the “spastic” vasoconstriction of the cerebral arteries due to a neuroinflammatory reaction [3]. Typically associated with complications following subarachnoid hemorrhage from aneurysm rupture, similar occurrences can also be observed in patients undergoing a tumor craniotomy (referred to as cerebral vasospasm after tumor craniotomy [CVACT]), as first reported in 1960 [4]. Despite diverse outcomes in cases reported from various countries, this condition is often underestimated. Unfortunately, the lack of widespread recognition of CVACT as a potential complication following tumor craniotomy results in delayed diagnosis and management, despite its significant mortality and morbidity rates. Although one study has suggested a predictive score for this condition through a retrospective study [5], the aim of our study is to comprehensively explore CVACT, covering the factors contributing to its occurrence, its mechanism, diagnostics, management, and outcomes.

## 2. Material and Methods

### 2.1. Study Design

This review has been reported following the PRISMA (Preferred Reporting Items for Systematic Reviews and Meta-Analyses) statement, as indicated in the PRISMA checklist [6]. The study protocol, including details of the methods used in the systematic review, has been deposited in the PROSPERO database registered with PROSPERO (https://www.crd.york.ac.uk/PROSPERO [accessed on 15 January 2025], no. CRD42024510503).

### 2.2. Search Strategy

A literature search was performed using relevant keywords to identify cases of “cerebral vasospasm” as a complication following tumor resection. A systematic search was performed using MeSH terms and keywords: (“Cerebral Vasospasm”[MeSH] OR “Intracranial Vasospasm”[MeSH]) AND (“Craniotomy”[MeSH] OR “Brain Tumor Resection”[MeSH]) AND (“Neurosurgery”[MeSH] OR “Surgical Procedures, Operative”[MeSH]). Additional keywords included “brain tumor”, “surgical complications”, and “postoperative vasospasm”. The search was conducted in PubMed and ScienceDirect, and additional cross-references were manually reviewed. Inclusion criteria in this study were case reports and case series, written in English, available in full-text. There were no restrictions on publication year. Gray literature was not excluded. The final search was completed in May 2024.

### 2.3. Study Selection and Eligibility Criteria

Duplicates were removed after completing the literature search. Based on titles and abstracts, primary selection was performed independently by two authors (KT and LBB). For the final selection, the full texts of the studies from the first selection were independently assessed for eligibility by two authors (KT and DYB). Any disagreements between the two authors were resolved through discussion with a third reviewer.

The inclusion criteria of this systematic review were all the articles that reported cerebral vasospasm following craniotomy tumor removal from case reports and or case series in English (full text) that contained enough details to allow a meaningful analysis, including demographic data, pathology, and location of the tumor, type of surgery and complication, vascular involvement, clinical manifestation and the onset of VS, diagnostic tools, management strategy, and outcome. Articles other than those with the above criteria were excluded.

### 2.4. Data Extraction

Two independent authors (KT and DYB) performed data extraction from all of the included studies into a pre-piloted data extraction form in Microsoft Excel. The third author (RAH) independently extracted data for validation. We extracted demographic data from each study, pathology, and location of the tumor, type of surgery and complication, vascular involvement, clinical manifestation and the onset of VS, diagnostic tools, management strategy, and outcome. The data were systematically reviewed and descriptively analyzed. We also performed a chi-square analysis to investigate the association between characteristics and vascular spasm.

### 2.5. Synthesis of Results

Descriptive statistics were used to calculate simple frequency, percentage, and proportion from the extracted data, reporting continuous data points as the median (IQR) or mean (±SD), categorical variables as percentages, and outcomes as a number and percentage.

### 2.6. Risk of Bias (RoB) Assessment and Quality of Evidence

Three authors (KT, DYB, and RAH) independently assessed the quality of the included studies using the Joanna Briggs Institute (JBI) critical appraisal checklist for case reports and series. Any discrepancies were resolved through discussion. In the absence of controlled studies, the quality of evidence was assessed through an examination of the reliability and validity of the public sources utilized in this review. Consideration was given to the relevance, reliability, and precision of these sources, and potential biases were discussed. Furthermore, the evaluation of evidence quality involved a comparative analysis of findings across diverse sources.

## 3. Results

### 3.1. Study Selection

The initial database and supplemental search yielded 196 articles. After 75 duplicates were removed, 121 articles were screened, and after titles and abstracts were screened, 73 articles were reviewed. Upon detailed evaluation,13 studies were excluded. Consequently, this systematic review included 16 case reports and 13 case series. The study selection process is presented in the flowchart (Figure 1).

### 3.2. Study Characteristics

We conducted a review of 27 articles, which comprised 14 case reports [7,8,9,10,11,12,13,14,15,16,17,18,19,20,21,22] and 13 case series [23,24,25,26,27,28,29,30,31,32,33,34], collectively detailing 60 cases of cerebral vasospasm following craniotomy for tumor (CVACT). The gender distribution was relatively balanced, with 33 patients identified as female (55%) and 27 as male (45%). The mean age of the patients was 44.05 years, and the median age was 47 years, with a range from 4 to 75 years. The most frequently identified tumor in this review was pituitary adenoma, found in 22 patients (36.66%), followed by meningioma, which accounted for 23.33%. All tumors analyzed were located at the skull base, in the sellar region in 43.33% of cases, with the middle cranial fossa being the predominant site (75%). Before surgery, patients typically presented with clinical symptoms indicative of increased intracranial pressure (ICP), mass effects, and localized symptoms. The most commonly utilized surgical technique was transsphenoidal surgery (50%). The characteristics of the studies included in this review are summarized in Table 1 and Appendix A.

Cerebral vasospasm typically manifests approximately 6.5 to 7 days postsurgery, with an observed range extending from 0 to 30 days. The symptoms are often indicative of a decline in the patient’s condition, accompanied by the emergence of new neurological signs. These manifestations can vary widely, ranging from asymptomatic cases to severe presentations that may include altered mental status in 41.66% of instances and progressive neurological deficits in 50%. A significant proportion of vasospasm cases, specifically 78.33%, is observed in the anterior circulation. Among the complications associated with postsurgical vasospasm, bleeding remains the most frequently reported, occurring in 53.33% of cases during the surgical procedure and in the postoperative phase [8,11,13,14,16,17,20,22,25,26,27,28,29,30,31,33,34,35]. The other complications reported included CSF leakage, with three cases occurring without additional complications [9,24,25] and four cases accompanied by other issues such as bleeding and two cases of meningitis [13,14,25,33]. In this study, imaging was determined to be the primary diagnostic tool, being employed in 40.96% of the cases. uTCD was utilized as a standalone diagnostic tool in only seven cases [24,28,29]. In other instances, uTCD was employed in conjunction with additional diagnostic modalities. However, the combination of imaging and uTCD monitoring yielded high survival rates and improved clinical outcomes.

The management strategies for cerebral vasospasm discussed in this review are consistent with those utilized for subarachnoid hemorrhage (SAH). The primary approach to treating vasospasm, as highlighted in the reviewed studies, involves pharmacological therapies such as nimodipine, levetiracetam, and papaverine, among others. This study reported a complete recovery rate of 56.75%. Further details regarding symptoms, diagnostic tools, and management strategies for vasospasm are available in Table 2 and Appendix A. In addition, we conducted a chi-square analysis, which revealed a significant correlation between tumor location and vascular vasospasm symptoms with the mortality rate (*p* = 0.015, r = 0.830; *p* = 0.02). This indicates a notable relationship between patient characteristics and the incidence of mortality following brain tumor surgery.

This review outlines several significant factors that are likely to contribute to the development of cerebral vasospasm. These factors include (a) the tumor’s placement within regions characterized by high vascularization, (b) the presence of abnormal vascular conditions resulting from narrowing or encasement, and (c) the accumulation of blood pools in the subarachnoid space and adjacent areas as a consequence of intraoperative bleeding.

### 3.3. ROB

According to the Risk of Bias in Visually Inspected Systematic Reviews (RoBVIS) assessment for case reports, six studies exhibited a high risk of bias [11,15,16,17,18,19], while the remaining studies demonstrated a low risk of bias [7,8,9,10,12,13,14,20,21,22]. The primary sources of bias were related to the description of adverse events, takeaway insights, and demographic details, as presented in Figure 2. Similarly, the RoBVIS assessment for case series indicated that six studies had a high risk of bias [24,26,27,28,29,30], while the other studies showed a low risk of bias [23,25,27,31,32,33,34,35]. Bias sources in the case series were attributed to factors such as clearly defined inclusion criteria, standardized measurement, consecutive sampling, complete inclusion, detailed demographic descriptions, and statistical analysis, as illustrated in Figure 2.

## 4. Discussion

Approximately 13–27% of complications following tumor craniotomy occur within 30 days postoperatively and are significant contributors to both morbidity and mortality. The predominant major complications observed include neurological issues, such as hematoma, cerebral edema, and seizures, as well as hemodynamic, respiratory, and metabolic complications. Intraoperative complications may include hemorrhage, cerebrospinal fluid (CSF) leakage, and nerve injury. Furthermore, the administration of anesthesia can also lead to complications following tumor craniotomy and other intracranial surgeries due to acute physiological changes that may occur during the recovery phase [36,37].

### 4.1. Patients’ Characteristics

Cerebral vasospasm after craniotomy tumor (CVACT) is a known phenomenon, yet it is not widely recognized due to its typical association with subarachnoid hemorrhage (SAH) [38,39]. The initial report on this condition was authored by Krayenbuehl et al. in 1960 [4], with several subsequent case reports expanding on the topic. This study systematically reviews all documented CVACT cases and thoroughly examines their characteristics.

A review of 60 cases, comprising both individual case reports and case series, suggests that CVACT is relatively rare; however, the true incidence remains unclear due to the absence of controlled studies. One study reported an incidence rate of 1.9% among 470 cases of postoperative cerebral vasospasm (CV) following brain tumor surgery [24]. Meanwhile, a recent retrospective study found a lower incidence of CVACT (called PoVS, postoperative vasospasm) of 0.61% among 2132 patients [5]. Our analysis of these 60 cases reveals that CVACT primarily affects young to middle-aged adults, with a mean age of 44.05 ± 16.8 years. Our result is in line with that study, but in our study, we did not impose an age limit, so we found the incidence of CVCT in pediatric cases. This result potentially reflects greater vascular elasticity in this age group. Additionally, gender does not seem to be a contributing factor, as our study found no significant difference between male (27 cases) and female (33 cases) patients. By contrast, aneurysmal subarachnoid hemorrhage (aSAH), often resulting from aneurysm rupture, has a higher prevalence among males aged 25 to 45, which may be linked to lower levels of sex hormone-binding globulin, thereby increasing the bioavailability of testosterone [40,41].

Cerebral vasospasm (CV) is a major cause of poorer outcomes and increased global disease burden in aneurysmal subarachnoid hemorrhage (aSAH). When CV after craniotomy tumor becomes symptomatic, morbidity and death rates are high. In our study, many patients did not recover from their major neurological deficits (34.24%) and the death rate was 25%. Given the high morbidity and mortality associated with this complication, once patients become symptomatic, identification of predisposing factors is key for a prompt diagnosis. This is very important in current clinical practice considering that early detection will increase the speed of response in providing management such that it can reduce the death rate and disability of sufferers.

### 4.2. Predisposing Factors

Based on this study, we identified several factors that may contribute to the occurrence of CVACT.

#### 4.2.1. Age

Our study identified that CVACT primarily occurs in younger and middle-aged individuals (mean age 44 years, range 4–75 years), consistent with previous research findings [5,23,40]. The highest incidence of CV in patients with aSAH was observed in the 50–57-year age group [41,42], contrasting with results from Malinova et al., who, through multivariate analysis, found the highest risk among individuals under 38 years [43]. Other studies have similarly observed that vasospasm in subarachnoid hemorrhage tends to occur at younger ages. This phenomenon is attributed to heightened vascular sensitivity and reactivity to vasodilators and vasoconstrictors in younger individuals. In contrast, with advancing age, the vascular response decreases, likely due to endothelial dysfunction involving endothelin-1 (ET1), a factor influenced by conditions such as hypertension, atherosclerosis, and chronic kidney disease, which contribute to vascular rigidity and a “spasmogenic” state [43,44]. Notably, smoking from a young age appears to provide some protection against vasospasm in later years. Smoking may promote ET1 production, decrease nitric oxide (NO) bioavailability, induce endothelial dysfunction, and accelerate atherosclerosis and loss of vascular elasticity, progressively impairing vasodilation. In cerebral vessels, smoking may lead to vessel narrowing and endothelial layer loss [43,45].

#### 4.2.2. Location and Characteristics of Tumors

The location and characteristics of the tumor can play a role in the occurrence of postcraniotomy vasospasm. Certain tumor types, particularly highly vascularized ones, and their proximity to vascular structures may elevate vasospasm risk. The skull base is a large, intricate region with essential anatomical structures, often making it susceptible to such complications. Tumors like adenomas and meningiomas tend to be larger and can readily extend into adjacent areas. Tumors in the medial cranial fossa are especially at risk for inducing vasospasm, as the internal carotid artery (ICA) and its branches, forming part of the circle of Willis, are located here. Growths extending into the sellar–suprasellar and lateral areas frequently cause arterial encasement or narrowing, especially within the anterior circulation. Solid tumors often lead to vascular compression and obstruction, while encapsulated tumors, with their capsules adherent to nearby arteries, increase the risk of vascular injury or perforation during removal. For instance, suprasellar masses commonly affect the A1 branch, whereas retrosellar tumors may involve the PcoA, P1, and basilar arteries [26,46,47].

In our study, all cases involved skull base tumors, with most in the middle cranial fossa (75%), demonstrating a significant correlation with CVACT (*p* = 0.015, r = 0.830). Our results are supported by previous studies where the incidence of PoVS also significantly occurred in tumors located in the middle cranial fossa (*p* = 0.01) in the parasellar region [5]. Among our cases, 26 tumors involved or extended into the sellar–suprasellar region [7,10,11,17,19,20,21,22,25,26,27,28,30,31,35]. Prior studies also report that CVACT in skull base tumors is most common in the sellar–suprasellar region, predominantly affecting pituitary adenomas and meningiomas, in agreement with our findings [48,49]. Though rare, posterior fossa tumors also have CVACT potential. A five-year retrospective review of posterior fossa tumors showed that 53.8% of patients with foramen magnum tumors had CVACT involving the vertebrobasilar artery [50]. Clinical symptoms of vasospasm here can range from asymptomatic to widespread vasospasm. In our study, only six patients had tumors in the posterior cranial fossa, and two cases tended to develop diffuse vasospasm [7,17,18,24,26,27]. Aoki et al. proposed that hypothalamic dysfunction could play a role in vasospasm among SAH patients, especially in cases of anterior skull base tumors where tumor growth or surgical intervention may affect the hypothalamus [23]. In our series, 17 out of 20 anterior skull base tumors demonstrated extension into the hypothalamic region.

#### 4.2.3. Vascular Factor

The circle of Willis (CoW) plays a critical role in ensuring cerebral perfusion by interconnecting the anterior and posterior circulatory systems through a series of anastomoses at the base of the brain. These structures guarantee blood supply to brain tissue through a series of anastomosis at the base of the brain, including the frontal, parietal, temporal, and occipital lobes, as well as the deepest structures in the brain such as the thalamus and basal ganglia. The circle of Willis (CoW) primarily consists of the ICA, which then bifurcates into the anterior cerebral artery (ACA), the middle cerebral artery (MCA), the posterior cerebral artery (PCA), and the anterior and posterior communicating arteries (AcomA and AComP). The MCA supplies the largest part of the cerebral hemispheres, including the expressive and receptive language centers, the basal ganglia, the posterior limb of the internal capsule, and the corona radiata [51]. Vasospasm involving the MCA can compromise perfusion to the lateral hemispheric regions, manifesting as motor deficits, aphasia, or neglect, depending on hemispheric dominance. Similarly, ACA vasospasm may lead to frontal lobe ischemia, characterized by executive dysfunction and lower extremity weakness. PCA vasospasm can result in occipital lobe ischemia, causing visual field deficits [51,52]. This study determined that vasospasm occurred in 47 cases (78.33%) in the anterior circulation of the circle of Willis, particularly in the internal carotid artery and its branches (see Table 1), with the most significant clinical symptoms associated with impairments in the brain regions supplied by the circle of Willis, specifically neurological deficits.

Vascular conditions significantly influence cardiovascular incidence. Vascular encasement or narrowing identified in preoperative circumstances in this study seems to be a pertinent risk factor to consider. While not all cases in this study indicated this, 38.33% of blood vessels that underwent encasement or constriction exhibited CVACT. This aligns with prior research [23]. Enhanced vascularization surrounding the tumor, present in some tumor forms, elevates the risk of intraoperative vascular damage, thus leading to blood spilling in the cistern.

Vascular factors influencing the occurrence of CVACT can be classified into two main categories: vascular damage due to surgical procedures, causing endothelial dysfunction, and the initiation of the inflammatory response. The surgical technique promptly causes harm to tissue and/or blood vessels, thereby initiating the release of inflammatory mediators and cytokines. In blood vessels exhibiting a positive response, norepinephrine, serotonin, and prostaglandins, which markedly increase during surgery as a physiological reaction to surgical stress, may rapidly precipitate cerebrovascular disease [40,41]. Moreover, surgical intervention may provoke stiffness in blood vessels due to relaxation mechanisms. In blood vessels exhibiting a tonal aberration, the incidence of spasms will be heightened. It is posited that direct and excessive vascular manipulation may interfere with the vasodilation process by obstructing the release of relaxant factors, hence shifting the balance of vascular tone towards spasmogenic states. This tonal imbalance may be linked to “reversible cerebral vasoconstriction syndrome”, described in instances of head trauma and neurosurgical interventions, although its mechanism is not sufficiently clarified [33,43,44]. Schiarti et al., through univariate analysis, confirmed this postulate statistically in a retrospective cohort where vascular manipulation intraoperative was significantly involved in CVACT [5].

#### 4.2.4. Surgical Factors

This section includes techniques and methods related to craniotomy, as well as the surgical difficulties and potential complications involved. The risk of cerebral vasospasm postcraniotomy can be directly influenced by the surgical techniques used in brain tumor surgery, particularly in procedures that involve tumors located in areas abundant in blood vessels or near vital vascular structures, as demonstrated in certain cases in our study. One of the techniques frequently linked to an elevated risk of vasospasm is brain tumor surgery in the suprasellar region or at the base of the brain, where significant vascular structures, such as the internal carotid artery or other large blood vessels, may be exposed or compressed [23,41,42]. Vascular spasms can be induced by direct manipulation of the blood vessels, even with the most meticulous microsurgical techniques.

Transsphenoidal surgery (TSS) is one of the most commonly used surgical techniques for removing tumors located in the sellar or suprasellar regions, such as pituitary adenomas. TSS is performed through transnasal (endoscopic endonasal endoscopic approaches—EEEA) and transsphenoidal approaches, which allow direct access to the tumor without opening the skull [15,23,53]. Although this technique has a high success rate and reduces the risk of injury to the brain or major blood vessels in the suprasellar region, the procedure still carries associated risks, including postoperative cerebral vasospasm [14,17,28,48]. A systematic review identified 34 cases of cerebral vasospasm occurring after TSS, with approximately 70.6% of vasospasm cases occurring with SAH [27]. In our study, 50% of cerebral vasospasm occurred postsurgery using the TSS technique and 26.66% during craniotomy. Moreover, the method of total resection or more aggressive tumor excision, intended to eradicate all tumor tissue, may result in damage to the affected blood vessels or compression of these arteries due to postoperative edema, consequently heightening the risk of vasospasm [29,35].

The results of our study found that all cases that developed cerebral vasospasm mostly experienced bleeding as the most common complication, both intraoperatively and postoperatively (53.33%). Another complication reported is CSF leakage with or without other complications (three cases), CSF leakage accompanied by bleeding (four cases), and CSF leakage accompanied by meningitis (two cases). Our investigation indicated that intraoperative hemorrhage was the primary contributor to vasospasm following tumor surgery in 14 patients, representing 23.33% of the cases. This surgical complication can lead to the accumulation of blood in the basal and subarachnoid cisterns, which may trigger vasospasm, much like what is observed in aneurysmal sub-arachnoid hemorrhage (aSAH) [4,23,42]. A significant number of patients experienced substantial intraoperative bleeding, as reported by Aoki et al. [22]. The stress associated with surgical manipulation can result in direct damage to blood vessels and mechanical irritation of the smooth muscle cells within these vessels or the vasa nervorum, due to excessive handling of the vascular structures. Furthermore, prolonged intraoperative hypotension and underlying atherosclerosis can elevate the risk of postoperative cerebral vasospasm [17,23].

### 4.3. Pathomechanism of Vascular Spasm

Cerebral vasospasm after tumor craniotomy (CVACT) can manifest shortly after surgery or may develop with a delay [20,27,34]. It can present as either localized or diffuse and may even occur without noticeable symptoms [24]. There is no clear mechanism that explains how cerebral vasospasm can occur after tumor craniotomy. The precise mechanism underlying cerebral vasospasm following a tumor craniotomy remains unclear. Although the pathomechanism of CVACT is not as well understood as that associated with vasospasm after subarachnoid hemorrhage (SAH), there are notable similarities and distinctions between the two conditions.

The release of blood into the subarachnoid space, which is typically the result of aneurysm rupture, is the most prevalent cause of SAH-induced vasospasm. The alleged involvement of blood products by SAH or ICH that triggered vasospasm was statistically strengthened in the Schiarti et al. study [5]. Endothelial injury and smooth muscle contraction in the blood vessels are the results of the release of vasoactive substances (such as hemoglobin) and inflammatory cytokines during the disintegration of red blood cells. Nitric oxide (NO), a potent vasodilator, is scavenged by hemoglobin from lysed red blood cells, resulting in impaired vasodilation. Endothelial dysfunction results in an imbalance between vasoconstrictors (e.g., endothelin-1) and vasodilators (e.g., NO). The prolonged contraction of the vessel walls is a result of the increased calcium influx that is induced by inflammatory processes in smooth muscle cells. Consequently, this extended vessel narrowing leads to delayed cerebral ischemia (DCI), which may result in infarction [17,37,54].

Conversely, CVACT is rarer and less researched than vasospasm in aSAH. The pathomechanism is not as clear-cut, given the absence of substantial blood leakage into the subarachnoid area, unlike in subarachnoid hemorrhage (SAH). Nonetheless, tumor manipulation and surgical trauma are considered to be significant factors. Direct surgical stress, tissue manipulation, and irritation during tumor excision might cause localized endothelial injury, facilitating vasospasm. Tumor excision may release inflammatory cytokines and chemical substances that lead to endothelial dysfunction, hence enhancing vasoconstriction, akin to observations in SAH [22,24,33]. Certain theories propose that tumors may induce vasospasm by releasing substances like lipid metabolites, modifying intracellular calcium levels through specific growth factors, or may be due to hypothalamic dysfunction, either directly from mechanical injury or indirectly via vasoactive substances and inflammatory biomarkers [16,54].

Tumor craniotomies can result in transient ischemia during the surgical procedure, particularly in eloquent or highly vascularized regions of the brain. This phenomenon may increase the likelihood of postoperative vasospasm. Furthermore, the increased brain edema that often follows tumor resection can exert additional pressure on cerebral vessels, thereby exacerbating the risk of vasospasm. Additionally, studies have shown that CVACT can lead to delayed cerebral ischemia that resembles the vasospasm associated with subarachnoid hemorrhage (SAH), with significant implications for morbidity and mortality.

In conclusion, SAH-vasospasm, blood breakdown products, and inflammation are primary triggers. In CVACT, surgical manipulation and trauma are the main culprits. However, further studies are required to confirm this hypothesis. Both types of vasospasm involve endothelial dysfunction, smooth muscle contraction, and a potential for ischemia; the trigger and mechanisms in CVACT are likely linked to mechanical and inflammatory factors from surgery rather than blood breakdown, as in SAH. Both, however, can result in significant neurological complications if not managed effectively.

### 4.4. Diagnosis of Vascular Spasm

The understanding of cerebral vasospasm following cerebrovascular accidents and cranial trauma (CVACT) events remains limited, which often results in delayed detection of associated complications. According to the findings of our study, alterations in consciousness and neurological conditions that progressively deteriorate within 0–7 days postoperatively may serve as initial indicators for suspecting CVACT (refer to Table 1). Angiography is recognized as the gold standard for the detection of cerebral vasospasm [23,28]. In our study, angiography was predominantly utilized as the diagnostic tool, applied in 34 patients, constituting 40.96% of the cases. Anderson et al. [55]. identified digital subtraction angiography (DSA) as the current standard method for the identification of vasospasm, providing accurate representations of intracranial vessels and the precise locations of arterial vasospasm. Their investigation involved 17 patients, revealing that 86% were diagnosed using both computed tomography angiography (CTA) and DSA, with a correlation coefficient of 0.757 (*p* < 0.001). Notably, the majority of patients in their study exhibited no vasospasm, while a limited number demonstrated severe spasm (131 vs. 7). Furthermore, CTA tends to provide a more accurate evaluation of spasm degree at proximal locations; however, it may both overestimate and underestimate the severity of spasm at distal sites. Generally, discrepancies between CTA and DSA primarily arise in the detection of mild and moderate degrees of spasm [55].

Transcranial Doppler ultrasonography (TCD) serves as a valuable, non-invasive tool for the diagnosis of cerebral vasospasm, measuring blood flow velocities within major intracranial arteries. It is frequently employed in critical care settings [31], particularly for patients suffering from subarachnoid hemorrhage [47], but has also proven valuable in detecting vasospasm after tumor craniotomy [56]. uTCD is utilized to assess blood flow velocities in the primary basal cerebral arteries. Bernoulli’s principle states that blood flow speed in an artery is inversely related to its diameter. During a vasospasm, as the artery narrows, blood flow speed increases. By comparing blood flow velocity with standard and baseline values, we can estimate changes in vessel diameter and detect vasospasms (mean flow velocities, MFV in cm/s). A recent review analyzed the use of TCD to diagnose vasospasm in patients with aneurysmal subarachnoid hemorrhage (aSAH). The findings show that uTCD for MCA has a pooled sensitivity of 81.5% and a pooled specificity of 96.6%. Additionally, it has a positive predictive value (PPV) of 93.7% and a negative predictive value (NPV) of 53.4% [57]. It is essential to recognize that uTCD demonstrates limited sensitivity in detecting vasospasm of the anterior cerebral artery (ACA). Furthermore, patients with aneurysms of the anterior communicating artery (AcomA) are particularly susceptible to false-negative results. This limitation arises because uTCD reliably indicates vasospasm only when there is a minimum 50% increase in mean flow velocity (MFV) over 24 h or when the MFV reaches 50 cm/s [46].

Cerebral vasospasm is diagnosed via uTCD examination when the following criteria are met: (1). The mean flow velocity (MFV) in MCA is ≥120 cm/s, or the Lindegard ratio (Ldg ratio) is >3. (2). The MFV in the basilar artery (BA) is ≥70 cm/s, or the Ldg ratio for the BA is ≥2. These parameters are critical for identifying potential vasospasm in patients [51,52]. The MCA, ACA, and posterior cerebral artery (PCA) can be examined through the transtemporal window or transforaminal window [46,57]. The basilar artery (BA) is typically evaluated using the transforaminal approach. An increased MFV finding in uTCD, combined with a low hyperemia index (HI < 1.2), may be correlated with vasospasm and delayed cerebral ischemia (DCI [58]. In this study, the majority of cases utilizing uTCD for diagnosing CVACT showed increased flow velocity, predominantly in the MCA, with or without the Lindegard ratio (Appendix A).

Systematic reviews affirm the diagnostic effectiveness of uTCD for identifying cerebral vasospasm, particularly highlighting its superior capability in detecting MCA vasospasm [50,53]. Unfortunately, in the current study, despite observing that vasospasm predominantly occurred in the anterior circulation (78.33%), only a limited number of cases (seven) utilized TCD as a primary diagnostic tool for CVACT [24,28,29]. This scarcity may be attributed to the retrospective nature of the included cases, as the use of ultrasonography may not have been widespread in the previous decade. Nevertheless, it is important to acknowledge the limitations of TCD, including its operator dependence and reduced efficacy in assessing distal branches of the cerebral vasculature or regions with suboptimal acoustic windows, such as those obscured by bone. Furthermore, the possibility of false-positive diagnoses should be considered, as elevated blood flow velocities may also arise from conditions such as hyperperfusion or increased cardiac output [26,33].

### 4.5. Management of Vascular Spasm

In the management of vasospasm complicating pituitary surgery, a range of therapeutic strategies can be utilized, akin to those employed in vasospasm associated with aneurysmal subarachnoid hemorrhage (SAH). Our study indicated that “Triple-H” therapy was the primary approach used in most cases, often in combination with nimodipine. This was typically followed by the intra-arterial administration of antispasmodic agents such as verapamil or papaverine, with mechanical angioplasty applied when warranted. “Triple-H” therapy (hypertension, hypervolemia, and hemodilution) yielded favorable outcomes in 34 out of 60 patients. The goal of this therapy is to enhance cerebral blood flow by increasing both blood pressure and cerebral perfusion, thereby alleviating the ischemic effects associated with vasospasm. Hyperdynamic therapy, which raises blood pressure through the use of vasopressors, is employed to maintain adequate perfusion of brain tissues. However, in certain instances, conservative management has been associated with less favorable outcomes [7,9,10,19,21,24,25,47]. However, the current concept of 3H therapy is no longer acceptable due to the lack of clinically supportive evidence that it can improve neurological outcomes in patients with vasospasm on SAH. The concept of hypervolemia, which aims to increase cerebral blood flow (CBF), actually worsens the outcome due to complications that occur due to the positive consequences of fluid balance. Therefore, maintaining euvolemic conditions with the administration of isotonic crystalloids (15 mL/kg in 1 h) is more widely recommended. Likewise, the concept of hemodilution can reduce oxygen delivery to brain tissue so that it is able to increase the cerebral ischemic volume. Therefore, hypervolemia and hemodilution do not provide benefits either for inducing hypertension or for improving cerebral perfusion pressure (CPP) in patients with SAH, but instead show a number of complications that contribute to the mortality and morbidity associated with vasospasm. Therefore, of the three components of 3H therapy, inducing hypertension is the only acceptable concept [52,59].

In our findings, nimodipine emerged as the preferred pharmacological treatment. According to Hao et al. [60], nimodipine significantly decreases mortality and the incidence of vascular vasospasm in patients with SAH. They strongly advocate for the early administration of nimodipine, especially in individuals under 50, to achieve optimal clinical outcomes, whether through oral or direct endovascular administration [23,55]. Nimodipine has also been used as a prophylactic drug for vasospasm in both adults and children [61,62]. In cases where pharmacological interventions prove ineffective, or depending on the clinical scenario, angiography can play both diagnostic and therapeutic roles. During angiography, intra-arterial vasodilators such as verapamil or nimodipine can be delivered directly to alleviate vasospasm. Additionally, balloon angioplasty is available as a method for dilating affected vessels, though it is generally reserved for more severe cases [12,23,25,32,33]. Levetiracetam is a medication primarily utilized for the management of seizures. Its pharmacological profile is noteworthy, as it functions as an anticonvulsant and possesses neuroprotective properties. Furthermore, levetiracetam has been shown to mitigate the development of cerebral vasospasm in cases of subarachnoid hemorrhage (SAH) [63,64,65].

## 5. Limitations

While case reports and case series having been placed at the bottom level of clinical studies, randomized controlled trials cannot be performed on this issue, and retro-prospective observational studies are particularly limited; thus, our findings remain limited to case studies and case series. The few reports of this case arise from the poor knowledge and experience that vasospasm occurs, not only in subarachnoid hemorrhage (SAH), but also in postcraniotomy tumor (CVACT). We only include case reports and case series in our review due to the rarity of the cases. Therefore, there is inadequate statistical analysis to overview the relationship of each variable with the CVACT. Several studies included in our study also had a high risk of bias. Case reports and case series are also considered lower on the hierarchy of evidence due to their inherent limitations in study design. Case reports are considered Level 5 (the lowest level) in the hierarchy of evidence because they do not involve control groups or large sample sizes. Meanwhile, case series are usually classified as Level 4 evidence. They provide more information than a case report by describing outcomes in a group of patients, but they still lack the rigorous control and randomization found in higher-level studies like randomized controlled trials (RCTs). Therefore, further studies are required to confirm our findings. However, our study is the first systematic review of CVACT. This study should be continued as a further study for the specific factors that cause vasospasm after brain tumor surgery due to the purpose of making a scoring system as an evaluation tool for patients with tumor surgery.

## 6. Conclusions

Cerebral vasospasm is a well-established clinical condition that must be understood to prevent delays in diagnosis, which can significantly affect mortality and morbidity rates. After a tumor craniotomy, any progressive deterioration in neurological function and consciousness should immediately raise concerns about cerebral vasospasm. Clinicians must be aware of risk factors that can be assessed preoperatively. These include patient factors, tumor characteristics, and surgical conditions, as well as intraoperative factors related to surgical techniques and potential complications. Recognizing these risk factors serves as an essential alert for clinicians when a patient’s condition declines after surgery. TCD is the most effective diagnostic tool for the early detection of this complication. There is no significant difference in treatment outcomes between vasospasm following subarachnoid hemorrhage (aSAH) and other types of vasospasm. As such, nimodipine remains the first-line therapy and should be utilized prophylactically when risk factors are identified.

## Figures and Tables

**Figure 1 jcm-14-02415-f001:**
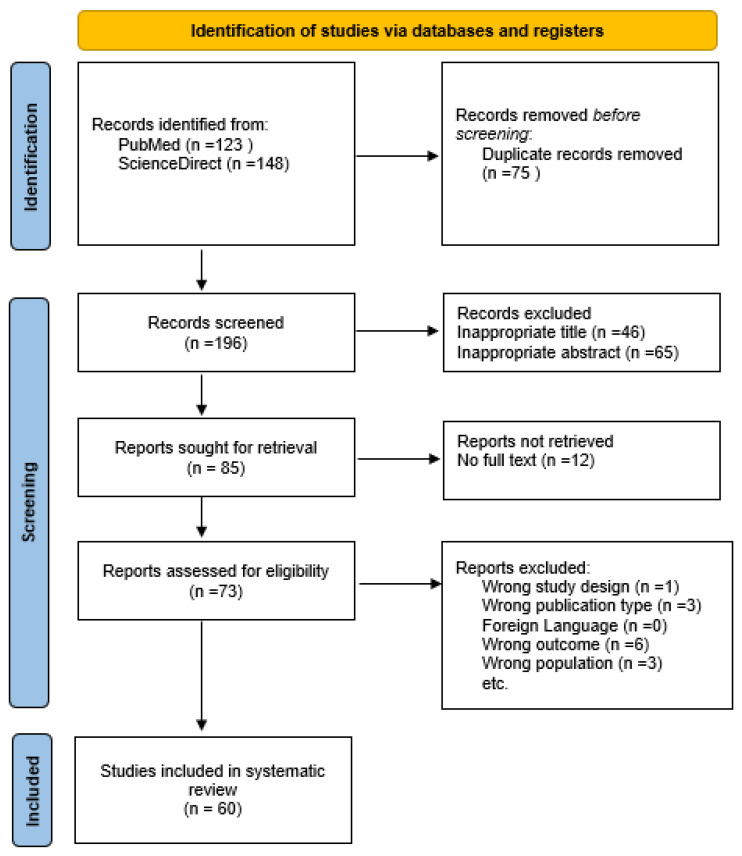
PRISMA Flowchart.

**Figure 2 jcm-14-02415-f002:**
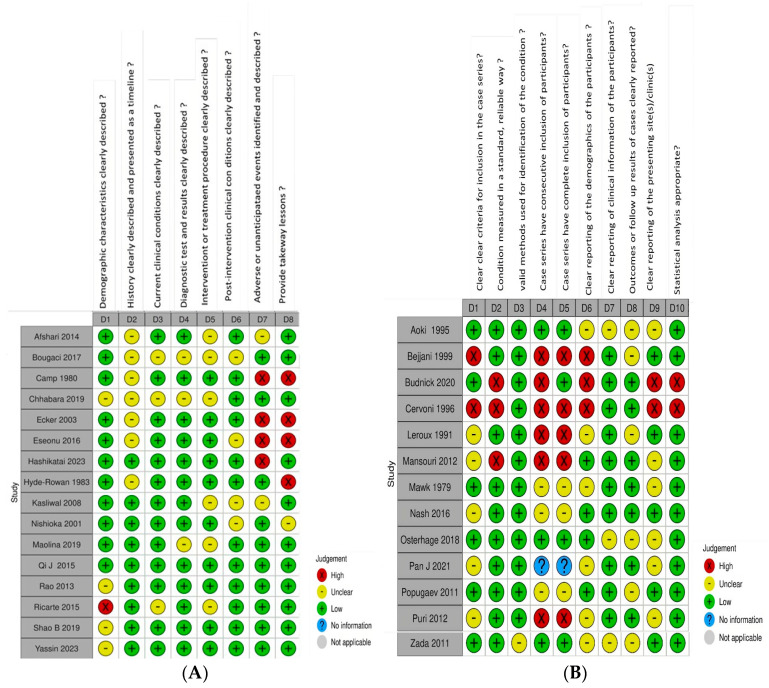
Risk of bias assessment for case reports [7,8,9,10,11,12,13,14,15,16,17,18,19,20,21,22] (**A**) and case series [23,24,25,26,27,28,29,30,31,32,33,34,35] (**B**).

**Table 1 jcm-14-02415-t001:** Characteristics and demographics of patients in the included studies.

Characteristics	Patients (n = 60)
Age (mean ± SD)	44.05 ± 16.8
Gender	
Female	33 (55%)
Male	27 (45%)
Location of tumor	
Anterior cranial fossa	7 (11.66%)
Middle cranial fossa	45 (75%)
Posterior cranial fossa	4 (6.66%)
Tumors extending into more than 1 fossae	4 (6.66%)
Type of Tumor	
Pituitary adenoma	22 (36.66%)
Meningioma	14 (23.33%)
Craniopharyngioma	7 (10%)
Large arachnoid cyst	1 (1.66%)
Ventricular colloid cyst	1 (1.66%)
Ruptured dermoid cyst	1 (1.66%)
Rathke’s cleft cyst	2 (3.33%)
Pleomorphic atypical aggressive adenoma	1 (1.66%)
Aggressive adrenocorticotropic	1 (1.66%)
Suprasellar mass	2 (3.33%)
Adenocarcinoma	2 (3.33%)
Chrdoma	1 (1.66%)
Schwannoma	3 (5%)
Neuroma acoustic	1 (1.66%)
Cavernous malformation	1 (1.66%)
Surgery Technique	
Transsphenoidal surgery	30 (50%)
Craniotomy	16 (26.66%)
Other	5 (8.33%)
NR	9 (15%)
Diagnostic Modality to CVACT	
Imaging	38 (40.96%)
uTCD	7 (11.66%)
Imaging + TCD	11 (18.33%)
Imaging + DSA	3 (5%)
uTCD + imaging + DSA	1 (1.66%)
Vascular-Affected CVACT	
Anterior circulation	47 (78.33%)
Posterior circulation	4 (6.66%)
Combination	9 (15%)
Vascular Factors (encasement, narrowing, displacement)	n = 23 (38.33%)
ICA	10 (43.47%)
ACA	5 (21.73%)
MCA	1 (4.34%)
VBA	2 (8.69%)
Combination	4 (17.39%)
ECA	1 (4.34%)
Clinical symptoms	
Neurological deficits (hemiparesis/plegi, visual change, etc.)	30 (50%)
Mental status changes	25 (41.66%)
Asymptomatic	1 (1.66%)
Other	4 (6.66%)
Complication of surgery	
Bleeding	32 (53.33%)
CSF leak	9 (15%)
Other	19 (31.66%)
Presence of blood postoperative (imaging tools) in the cisterna system	21 (35%)
The most frequently identified causative factors	
Vascular factors	23 (38.33%)
Presence of blood in the cisterna system	21 (35%)
Intraoperative bleeding	15 (5%)
Outcomes	
Survive	37 (61.66%)
Complete recovery	21 (56.75%)
Incomplete recovery	16 (34.24%)
Died	15 (25%)
NR	8 (13.33%)

ACA: Anterior Cerebral Artery, CSF: Cerebrospinal Fluid, DSA: Digital Subtraction Angiography, ECA: External Carotid Artery, ICA: Internal Carotid Artery, MCA: Middle Cerebral Artery, NR: Not Reported, SD: Standard Deviation, TCD: Transcranial Doppler, uTCD: Ultrasound Transcranial Doppler, VBA: Vertebrobasilar Artery. CVACT: Cerebral Vasospasm After Craniotomy Tumor.

**Table 2 jcm-14-02415-t002:** Characteristics of symptoms, diagnostic tools, management, and outcome of CV.

Characteristics	Variable
Onset of VS (time after surgery to vasospasm)	6.59 (0–30 days)
Clinical symptoms	n = 60
Neurological deficits (hemiparesis, visual change, etc.)	30 (50%)
Mental status changes	25 (41.66%)
Asymptomatic	1 (1.66%)
Other	4 (6.6%)
Diagnostic Tools	
uTCD	n = 7
Death	3 (42.85%)
Life	4 (57.14%)
Complete recovery	3 (75%)
Incomplete recovery	1 (25%)
Imaging	n = 38
Death	10 (26.31%)
Life	20 (52.63%)
Complete recovery	10 (50%)
Incomplete recovery	10 (50%)
NR	8 (21.05%)
Combination TCD-Imaging	n = 11
Death	1 (9.09%)
Life	10 (90.90%)
Complete recovery	7 (70%)
Incomplete recovery	3 (30%)
Therapy	
Pharmacology	
Nimodipin	32 (53.33%)
Levetiracetam	2 (3.33%)
Paperavine	3 (5%)
Atorvastatin	1 (1.66%)
Aminofilin	1 (1.66%)
Isoprel	1 (1.66%)
Nicardipine	1 (1.66%)
Verapamil	4 (6.66%)
Milrinone	1 (1.66%)
Noradrenaline	1 (1.66%)
Not described	4 (6.66%)
Supportive 3H	12 (24.07%)
Invasive intervention	
Intra-aortic Balloon Pump	1 (1.66%)
Balloon Angioplasty	5 (8.33%)
VP Shunt	2 (3.33%)
Angioplasty	2 (3.33%)
Lumbar LCS Drainage	1 (1.66%)
Angiography	1 (1.66%)
Not Described	39 (65%)
Combination	
3H + Pharmacology	16 (29.62%)
3H + Neurointervention	4 (7.40%)
Pharmacology + Neurointervention	5 (9.25%)
3H + Pharmacology + Neurointervention	4 (7.40%)

3H: Hemodilution, Hypertension, and Hypervolemia, LCS: Lumbar Cerebrospinal, NR: Not Reported, TCD: Transcranial Doppler, VP Shunt: Ventriculoperitoneal Shunt, VS: Vasospasm.

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
