# Peer review of "Cerebral Vasospasm as a Critical Yet Overlooked Complication Following Tumor Craniotomy: A Systematic Review of Case Reports and Case Series"

_jcm, 2025, doi:10.3390/jcm14072415_

Round 1
Reviewer 1 Report
Comments and Suggestions for Authors
In this systematic review, the authors aimed to study the incidence and factors associated with the development of cerebral vasospasm after tumor craniotomy. This is indeed an under-recognized and treatable cause of significant morbidity and mortality after tumor craniotomy. Certain concerns related to the results and interpretation of the data remain unaddressed:
1. Abstract - Conclusion: "Cerebral vasospasm (CV) after craniotomy tumor removal occurs generally in pituitary adenoma tumors located in the cranial fossa (middle cranial fossae) with complications of intraoperative bleeding involving the anterior cerebellar artery, especially the MCA." The anterior cerebellar artery does not constitute anterior circulation and is a branch of the basilar artery (post circulation). A middle cranial fossa surgery should not affect the anterior cerebellar artery as such. Please check and make the necessary corrections.
2. Page 4, Line 141 - "Among the complications associated with 141 post-surgical vasospasm, bleeding remains the most frequently reported, occurring in 25% of cases during the surgical procedure and in 43.33% in the postoperative phase." Bleeding, CSF leakage and meningitis cannot be reported as complications of post surgical vasospasm. However, these can definitely lead to the development of vasospasm. Please check and make necessary corrections. Please explain this statement if the authors disagree.
3. Page 4, Line 150 - "The primary approach to treating vasospasm, as highlighted in the reviewed studies, involves pharmacological therapies such as nimodipine, levetiracetam, and papaverine, among others." Levetiracetam has not been approved or used for the management of vasospasm. It is used to treat seizures related to vasospasm or prophylactically in aneurysmal subarachnoid hemorrhage. Please cite an appropriate reference to support this practice.
4. Table 2 - Bleeding and CSF leak have been reported as intraoperative and post-operative complication. Are these two events mutually exclusive? Based on the information provided, it implies that CSF leak occurred in a total of 9 patients collectively (6+3). Is that fair to assume? please clarify in the manuscript.
5. Diagnostic imaging - as per the information provided, TCD was used to diagnose vasospasm and is recommended by authors in the conclusion for early diagnosis of post-craniotomy vasospasm. What TCD criteria were used to diagnose vasospasm? Was it universal and consistent across all the studies that the authors included in the analysis. Please clarify?
6. Triple H therapy was used in 12 patients as per the manuscript. Triple H therapy has been associated with increased mortality for the management of symptomatic vasospasm in aneurysmal subarachnoid hemorrhage. Do the authors recommend using triple H therapy based on their review of studies? Please clarify.
7. An important limitation of this study is the small sample size or number of patients and low-quality studies (case reports and case series) that have been included in the final analysis. Please discuss this as a limitation in the discussion.
Comments on the Quality of English LanguageThere are multiple grammatical errors throughout the manuscript. The percentages quoted in Table 2 need to be corrected - 6,66% should be changed to 6.66%, at numerous places in the table. Could you check the decimals for quoting percentages? Please proofread the article or use online grammar check software to make sure the errors have been corrected.
Author Response
- Abstract - Conclusion: "Cerebral vasospasm (CV) after craniotomy tumor removal occurs generally in pituitary adenoma tumors located in the cranial fossa (middle cranial fossae) with complications of intraoperative bleeding involving the anterior cerebellar artery, especially the MCA." The anterior cerebellar artery does not constitute anterior circulation and is a branch of the basilar artery (post circulation). A middle cranial fossa surgery should not affect the anterior cerebellar artery as such. Please check and make the necessary corrections.
RESPONSE: Thank you for your input. We have corrected the sentence of the conclusion statement in the abstract according to the data we got.
- Page 4, Line 141 - "Among the complications associated with 141 post-surgical vasospasm, bleeding remains the most frequently reported, occurring in 25% of cases during the surgical procedure and in 43.33% in the postoperative phase." Bleeding, CSF leakage and meningitis cannot be reported as complications of post-surgical vasospasm. However, these can definitely lead to the development of vasospasm. Please check and make necessary corrections. Please explain this statement if the authors disagree.
RESPONSE: Thank you for your input. We agree with the reviewer's statement. Because the data in Table 1 row 141 does show intraoperative and postoperative complications that could contribute to the occurrence of vasospasm.
- Page 4, Line 150 - "The primary approach to treating vasospasm, as highlighted in the reviewed studies, involves pharmacological therapies such as nimodipine, levetiracetam, and papaverine, among others." Levetiracetam has not been approved or used for the management of vasospasm. It is used to treat seizures related to vasospasm or prophylactically in aneurysmal subarachnoid hemorrhage. Please cite an appropriate reference to support this practice.
RESPONSE: Thank you for your input. We agree with reviewers that levetiracetam is generally used as an anti-seizure. However, studies have found that levetiracetam can reduce vasospasm that occurs in traumatic SAH and TBI (Wang, 2006; Kirmani, 2014; Amano,2023)
- Table 2 - Bleeding and CSF leak have been reported as intraoperative and post-operative complication. Are these two events mutually exclusive? Based on the information provided, it implies that CSF leak occurred in a total of 9 patients collectively (6+3). Is that fair to assume? please clarify in the manuscript.
RESPONSE: Thank you for your input. We have explained in the results section of lines 154-158
- Diagnostic imaging - as per the information provided, TCD was used to diagnose vasospasm and is recommended by authors in the conclusion for early diagnosis of post-craniotomy vasospasm. What TCD criteria were used to diagnose vasospasm? Was it universal and consistent across all the studies that the authors included in the analysis. Please clarify?
RESPONSE: Thank you for your input. We will outline the CV diagnostic criteria in the diagnostics of vasopasm section. We have added the TCD results of the primary studies in suplementary 2.
- Triple H therapy was used in 12 patients as per the manuscript. Triple H therapy has been associated with increased mortality for the management of symptomatic vasospasm in aneurysmal subarachnoid hemorrhage. Do the authors recommend using triple H therapy based on their review of studies? Please clarify.
RESPONSE: Thank you for your input. We will discuss our opinions and recommendations in the discussion section (line 477-488)
- An important limitation of this study is the small sample size or number of patients and low-quality studies (case reports and case series) that have been included in the final analysis. Please discuss this as a limitation in the discussion.
RESPONSE: Thank you for your input. We will discuss the limitations of our study in the discussion section.
- There are multiple grammatical errors throughout the manuscript. The percentages quoted in Table 2 need to be corrected - 6,66% should be changed to 6.66%, at numerous places in the table. Could you check the decimals for quoting percentages? Please proofread the article or use online grammar check software to make sure the errors have been corrected.
RESPONSE: Thank you for your input. We will check and correct grammatical errors in our manuscript again.
Reviewer 2 Report
Comments and Suggestions for Authors
The aim of the authors is to provide an insight into vasospasm after surgery ofr intracranial tumors. This is an interesting topic.
However, several limitations are present in this manuscript. Just to mention a few, the authors did not include in their research the term "Delayed cerebral ischemia -DCI" which could have substantially reduced the amount of works considered.
Specific mention on the outcomes and the brsin territory involved by vasospasm should be better discussed. Also, when searching for statistical significance, a correction for multiple comparisons should be provided.
Finally, the work is a bit confused: why including a part of the discussion in the results section? (see paragraph 3.1).
Comments on the Quality of English LanguageSome anatomical terms should be corrected (for instance "cranial fossae") and english editing is encouraged
Author Response
- However, several limitations are present in this manuscript. Just to mention a few, the authors did not include in their research the term "Delayed cerebral ischemia -DCI" which could have substantially reduced the amount of works considered.
RESPONSE: Thank you for your input. We recognize that one of the outcomes of vasospasm is delayed cerebral ischemia (DCI). However, our article does not specifically address the complications associated with vasospasm. The primary objective of our study is to explore the mechanisms and specific factors contributing to the incidence of vasospasm following tumor craniotomy. We are particularly interested in whether there are unique characteristics in the pathomechanism that set it apart from the well-established vasospasm associated with subarachnoid hemorrhage (SAH). Consequently, the studies included in our article focus exclusively on cases of vasospasm, without examining the resulting complications.
- Specific mention on the outcomes and the brsin territory involved by vasospasm should be better discussed. Also, when searching for statistical significance, a correction for multiple comparisons should be provided.
RESPONSE: Thank you for your input. We have added in the vascular factors discussion (line 291-306)
- Finally, the work is a bit confused: why including a part of the discussion in the results section? (see paragraph 3.1).
RESPONSE: Thank you for your input. We will improve the paragraph by focusing on conveying the results of the study.
- Some anatomical terms should be corrected (for instance "cranial fossae") and english editing is encouraged.
RESPONSE: Thank you for your input. We will correct the terms and spelling in our manuscript, and we will editing our english.